# Magnetic Resonance Spectroscopy in Diagnosis and Follow-Up of Gliomas: State-of-the-Art

**DOI:** 10.3390/cancers14133197

**Published:** 2022-06-29

**Authors:** Malik Galijasevic, Ruth Steiger, Stephanie Mangesius, Julian Mangesius, Johannes Kerschbaumer, Christian Franz Freyschlag, Nadja Gruber, Tanja Janjic, Elke Ruth Gizewski, Astrid Ellen Grams

**Affiliations:** 1Department of Neuroradiology, Medical University of Innsbruck, 6020 Innsbruck, Austria; malik.galijasevic@i-med.ac.at (M.G.); ruth.steiger@i-med.ac.at (R.S.); tanja.janjic@i-med.ac.at (T.J.); elke.gizewski@i-med.ac.at (E.R.G.); astrid.grams@i-med.ac.at (A.E.G.); 2Neuroimaging Research Core Facility, Medical University of Innsbruck, 6020 Innsbruck, Austria; 3Department of Radiation Oncology, Medical University of Innsbruck, 6020 Innsbruck, Austria; julian.mangesius@i-med.ac.at; 4Department of Neurosurgery, Medical University of Innsbruck, 6020 Innsbruck, Austria; j.kerschbaumer@i-med.ac.at (J.K.); christian.freyschlag@i-med.ac.at (C.F.F.); 5VASCage-Research Centre on Vascular Ageing and Stroke, 6020 Innsbruck, Austria; nadja.gruber@uibk.ac.at; 6Department of Applied Mathematics, University of Innsbruck, 6020 Innsbruck, Austria

**Keywords:** magnetic resonance spectroscopy, glioma, imaging biomarkers

## Abstract

**Simple Summary:**

Magnetic resonance spectroscopy (MRS) is a useful technique in diagnosis and follow-up of gliomas. In this review we provide an insight in the use of both proton and phosphorous MRS in clinical and scientific every day practice.

**Abstract:**

Preoperative grade prediction is important in diagnostics of glioma. Even more important can be follow-up after chemotherapy and radiotherapy of high grade gliomas. In this review we provide an overview of MR-spectroscopy (MRS), technical aspects, and different clinical scenarios in the diagnostics and follow-up of gliomas in pediatric and adult populations. Furthermore, we provide a recap of the current research utility and possible future strategies regarding proton- and phosphorous-MRS in glioma research.

## 1. Introduction

Correct and timely diagnosis of a glioma is extremely important. Patients benefit from an early diagnosis and precise follow up. Early diagnosis can have an impact on quality of life and overall prognosis [1,2], and the importance of correct follow-up is well known in all tumor types and subtypes. With an increase in our knowledge of glioma pathology, conventional MRI is struggling in our experience especially in follow-up and distinguishing radiation necrosis (RN), pseudoprogression (PsP), and tumor progression (TP) in high grade gliomas. Besides regular structural sequences, in the diagnosis and follow-up of these tumors, additional sequences are used to provide further information about the pathological processes in gliomas [3], or the standard sequences are being quantified in order to differentiate between different pathological tumor types [4]. To this end, in our centre we routinely use perfusion weighted imaging (PWI), delayed contrast-enhanced imaging, and proton-MRS.

MRS is a special magnetic resonance technique used to quantify different metabolites in a voxel (volume pixel) of tissue. So far in clinical practice only hydrogen (1H), or proton-MRS is used, although an MRS spectrum can be obtained from any element with non-zero spin. MRS with other chemical elements has been used in research purposes (most commonly phosphorous—31P and carbon—13C). The main metabolites in 1H-MRS in clinical settings are N-acetylaspartate (NAA), choline (Cho), and creatine (Cr). It is a well known technique used in clinical practice in the diagnosis and follow up of various brain lesions [5,6]. The main metabolites in 31P-MRS are phosphocreatine (PCr), adenosine triphosphate (ATP), inorganic phosphate (Pi), phosphomonoesters (PME) and phosphodiesters (PDE). In addition, intracellular pH and magnesium levels can be calculated.

As an example of tumor protocol MRI, patients with brain tumors can be imaged using sequences described in the consensus recommendations by Ellingson et al. [7]. If the needed hardware and software allow, these basic sequences can be supplemented with some of the more advanced techniques in order to better discriminate between various subtypes of tumors, as described by Malik et al. [8]. In our center the patients with brain tumors are imaged with 3 T and 1.5 T machines with the following sequences: axial T2-turbo spin echo (TSE), 3D fluid-attenuated inversion recovery (FLAIR), axial diffusion weighted imaging (DWI), axial susceptibility-weighted imaging (SWI) and 3D T1 magnetization prepared—rapid gradient echo (MPRAGE) before and after contrast application. Besides these “classical” sequences, we use PWI, MRS, and delayed post-contrast imaging in order to predict pathological diagnosis, monitor answer to therapy, radiation changes, and possible recurrence of a disease. If the tumor is localised in proximity to the eloquent cortical centers, we use functional magnetic resonance imaging (fMRI) with motor and language paradigms to access possible infiltration and to plan surgical approach (only on 3 T scanners).

In light of a recent publication by the World Health Organization (WHO) Classification of Tumors of Central Nervous System in 2021, in addition to significant changes in the classification of gliomas, it is important to provide an update regarding the place of MRS in the diagnostics, follow up, and research of gliomas. Since it is based on molecular features, the WHO Classification of Tumors of Central Nervous System (2021) introduced many changes for glial brain tumors. Thus, it is of outmost importance to review the value of commonly available 1H-MRS and the novel 31P-MRS in both classifying the tumor subtype before resection and distinguishing RN, PsP and TP during follow-up. There is already a great number of reviews and original research papers dealing with MRS in gliomas, however, to the best of our knowledge this is the first review analysing this topic regarding the new WHO classification. The aim of this review is to provide an overview of the clinical state-of-the-art of this rapidly evolving technology in the diagnostic work up of glioma patients by means of MRS, and to further contribute to the planning of future research studies by enabling better preoperative and postoperative radiological assessment of gliomas in light of the new WHO-classification. While MRS in the adult population has been extensively explored in the literature, little has been described about its value in the pediatric population. For this reason, a comparison of MRS examination and interpretation between adult and pediatric patients in the context of neuro-oncology assessment is presented here.

## 2. Classification of Gliomas

In 2016 the WHO for the first time included molecular parameters in the final diagnosis of tumors of the central nervous system [9]. This was underlined in a new version of the classification in 2021, in which some of these molecular markers were made even more important than the histological appearance. Furthermore, some new tumor types and subtypes were introduced [10].

Besides purely didactic changes like using Arabic instead of Roman numerals in the designation of grade, or using “type” and “subtype” instead of “entity” and “variant”, the new classification also brought some more clinically important changes. As previously mentioned, in some cases the molecular features supersede histologic characteristics. For example, histologically “low-grade” astrocytoma, IDH-wildtype (IDH-wt) and EGFR-amplification, TERT-promotor mutation, or combined gain of chromosome 7 and loss of chromosome 10, can be considered as glioblastoma (GBM), and consequently as WHO Grade 4 [10]. The new classification of gliomas with the most important molecular features is given in Table 1. Some of the tumor types are newly recognised, and have not yet been given a WHO grade.

## 3. Technical Overview

### 3.1. Pediatric Population

Newborns until up to 3–4 months can be examined by the feed and wrap technique using special ear cuffs and a vacuum pillow without sedation but with constant monitoring of oxygen saturation by a neonatologist. For older babies and young children, until about reaching the school age, MRI examinations usually have to be acquired under general anesthetic, certainly including the monitoring of vital functions.

### 3.2. Older Children and Adults

For older children and adults (under the condition of compliance and physical feasibility) the MR spectroscopy measurement procedure is identical to structural imaging of the brain. Ear protection for noise reduction is mandatory for all age cohorts and all MRI field strengths. In the next two subsections an example of standardized spectroscopy sequence planning is given.

### 3.3. Sequence Planning—1H-MRS

Planning of the spectroscopy sequence (chemical shift imaging—CSI vs. single voxel spectroscopy—SVS) particularly depends on the location of the brain region under investigation and the medical purposes. Shimming for routine patients is performed automatically, whereas, for study participants and scientific investigations, it is carried out manually in order to avoid line broadening of the spectral width at the half amplitude of the signal (FWHM) at physical values for 1.5 T: SVS < 13 Hz, CSI < 15 Hz and at 3 T: SVS < 20 Hz, CSI < 25 Hz [31].

### 3.4. Sequence Planning—31P-MRS

31-P-MRS is performed in a 3 T MRI machine using a double-tuned 1H/31P volume head coil (Rapid Biomedical, Würzburg, Germany). The sequence is planned on an isotropic T2-weighted 3D sequence. Boundary regions, as well as regions filled with air or bone are spared, to avoid voxel contamination. The measurements are acquired using the parameters given in Table 2, and based on a conventional sequence by Siemens and described by Hattingen et al., in order to ensure reproducibility and applicability for potential future clinical use [32,33,34].

## 4. Diagnosis and 1H-MRS

### 4.1. Pediatric Population

The most common glioma in the pediatric population is pilocytic astrocytoma [35]. This tumor type is not only the most common glioma, but is the most common brain tumor in children in general, accounting for around 15% of all brain tumors in this population [36]. The imaging of the pilocytic astrocytoma is fairly straightforward using regular structural MRI sequences, however, when performed, MRS show a high Cho/NAA and Cho/Cr ratio, and low Cr with a decreased NAA/Cr ratio [37]. These changes can be seen in an example of a 2 year old patient with histologically proven cerebellar pilocitic astrocytoma (Figure 1).

High-grade gliomas, not otherwise specified (NOS) are the most common high-grade gliomas in children [35]. There is not much difference in interpreting spectroscopy findings in children compared to those of adults. The best indicator of malignancy was found to be the NAA/Cho ratio. High-grade astrocytic tumors tend to have more decreased NAA and increased Cho compared to the lower-grade tumors [38]. However, as can be appreciated in our real-world example, relying solely on a spectroscopy can be misleading, especially in differentiating various diffuse tumors (as in our example in Figure 2), as both, lower and higher grade diffuse tumors can have similar spectra, albeit a difference between diffuse and circumscribed gliomas is usually clear (Figure 1 vs. Figure 2).

### 4.2. Adult Population

#### 4.2.1. Low-Grade vs. High-Grade

Preoperative grading of gliomas can be difficult based on conventional MRI, especially with the absence of significant edema and/or contrast-enhancement. This is a common diagnostic challenge, even in high-grade tumors. In the case of diagnostic uncertainty, additional information can be obtained with MRS [39,40].

In gliomas, the general rule is that the NAA and Cr decrease, and other metabolites increase with higher tumor grades. Some authors suggest using a Cho/NAA ratio greater than 2.2 to predict higher grade, and a presence of myoinositol to predict the lower grade lesions [41,42]. However, as shown in Figure 3 and in our own experience, low-grade gliomas can also have similar spectra as high-grades (but rarely vice-versa).

#### 4.2.2. Follow-Up and MRS

MRS is important in postoperative follow up, especially in high grade gliomas. As previously described, MRS is used as a valuable tool in the diagnosis and follow-up of gliomas [43]. Both residual tumor or tumor recurrence may be observed after brain irradiation. Furthermore, PsP and RN constitute two further different types of adverse therapeutic effects.

Radionecrosis is determined as the development of necrotic brain tissue after irradiation, which emerges between three months and one year after radiotherapy [22] and affects about 20% of radiotherapy in GBM patients [44] especially after receiving higher radiation doses [45] and additional chemotherapy [46].

PsP is defined as a transient and self-limited volume increase without evidence of vital tumor, which occurs between two and five months after the initiation of radiation, and affects approximately 20% of patients with concomitant chemo-radiation. It is assumed to be a mixed effect of treatment reaction and a collapse of the blood-brain barrier [44]. As differentiation from recurrent tumor is difficult, a close monitoring with frequent MRI is recommended so as not to misinterpret an RN and PsP [47].

Due to the expected high rate of treatment effects, and the fact that each of the three above-mentioned post therapeutic conditions require different therapeutic strategies [44], monitoring is crucial.

Failing to differentiate radiation induced changes or drug-induced PsP from TP can have dire clinical consequences.

The effect of radiotherapy on the brain are early alterations in metabolic activity, which finally result in tissue degradation, yet antecede the development of symptoms and occur before evidence of changes can be determined on structural images using conventional MRI in early post treatment scans [48].

The differentiation between PsP and TP with MRS poses a major diagnostic challenge, particularly with the use of single-voxel acquisitions, as discussed herein. Both types of lesions can demonstrate neuronal loss/dysfunction (decrease of NAA), abnormal cellular membrane attenuation/integrity and proliferation (increase of Cho), and anaerobic metabolism (high Lac/lipid ratio). An increased Cho/NAA aand Cho/Cr ratio corresponds with tumor recurrence [49,50,51,52,53,54]. PsP could be diagnosed based on elevated lipid signals on MRS [55]. However, these effects may not always be seen, as the absence of Cho or a low Cho/NAA ratio has also been observed. Contrarily, patients with TP present with lower lipid signals and a high Cho/NAA ratio. The evidence of elevated lipid signals, together with a low Cho/NAA ratio, may help to distinguish PsP from TP [52].

Consequently, the most important metabolite in differentiating TP from radiation-induced PsP is choline, as disturbances in the biosynthesis of cell membranes and metabolic turnover are reflected by an increase in choline. As in a primary tumor, choline will be increased in TP, whereas it will be decreased in RN and radiation induced PsP, together with NAA and creatine. In our experience, the most common clinical encounter is with patients that have imaging characteristics of both TP and radiation-induced PsP (Figure 4).

In radionecrosis, progressive metabolic changes induce a decreased concentration of the neuronal marker N acetylaspartate (NAA), which reflects cell death by apoptosis or neuronal dysfunction. Disturbed biosynthesis of cell membranes and metabolic turnover are reflected by an increase in choline (Cho). However, creatinine (Cr), which constitutes the marker of energy metabolism, is considered to be unaffected by radiation damage. Therefore, in brain tissue developing radionecrosis increased Cho/Cr ratio is observed [6,53,54,56,57,58].

A meta-analysis involving 1174 patients treated for GBM could show that using advanced MRI techniques leads to greater diagnostic accuracy when compared to using only conventional MRI for follow-up of response to treatment, leading to greater sensitivity and specificity [59]. It has recently been suggested [60] that for the differentiation of PsP, RN and TP, DWI and PWI should be performed after conventional sequences. However, in the case of remaining diagnostic uncertainty, MRS was shown to provide complementary information. Some studies described cut-off values for differential diagnosis.

For differentiating between PsP and TP, the following MRS ratios and cut-offs were suggested: a Cho/NAA ratio under 1.47–2.11 and Cho/Cr ration under 0.82–2.25 indicated PsP. Cho/NAA ratio had a mean of 2.72 for TP, and 1.46 for RN (*p* < 0.01) [61]. Figure 4 shows the Cho/NAA ratio under cut-off for histologically confirmed GBM progression.

Ultimately, several advanced MRI modalities, including MRS, seem to improve differentiation between PsP, RN and TP, compared with conventional MRI.

## 5. Future Aspects and 31P-MRS

1H-MRS is a well established technique in clinical practice for the diagnosis and follow-up of glioma. As previously discussed, depicting the various 1H-metabolites can bring the necessary information for classifying a brain lesion, and especially to monitor the answer to therapy and possible recurrence. Recently however, many research groups are trying to find a similar clinical place for 31P-MRS. It showed potential to further classify the tumors, and even predicting recurrence.

Future research strategies should focus on differences in MRS in tumors with various molecular footprints. In a recently published study, differences in energy metabolites of GBMs with variations in MGMT and EGFR status were shown. These results showed indications of faster cell reproduction in MGMT-methylated and EGFR-amplified tumors and higher apoptotic activity in EGFR-amplified tumors regardless of the MGMT-status [62]. Another study showed lower lactate levels and intracellular pH in IDH-mutant gliomas compared to IDH-wild type gliomas [63]. However, other 31P-MRS markers were not significantly altered and could not predict the IDH-mutation status [64]. The same group also showed the ability of 31P-MRS to predict the site of progression of GBM under angiogenic therapy. Namely, the elevated intracellular pH was regarded as a predictor for a progression of recurrent GBM treated with bevacizumab [65].

It would be interesting to further examine the possibilities of both 1H and 31P MRS in preoperative assesement of different molecular markers and the possibility of this method to predict progression. Regarding the 2021 WHO Classification of CNS Tumors these methods could be helpful in distinguishing various subtypes of gliomas.

As already discussed, spectroscopy can also be a reasonable tool in distinguishing various changes in tumor metabolism after standard therapy regimen with chemotherapy and radiation. In a recent paper, differences in energy metabolism between tumors in various stages according to RANO criteria were used using 31P-MRS. Among other results, in progressive disease patients, normalisation of energy metabolites after the induction of therapy was seen [66]. In another study, among other results, regional differences between normal appearing brain and various tumor areas in patients with GBM were shown, also using 31P MRS. Contrast-enhancing areas had increased intracellular pH and magnesium levels, decreasing with the distance from the tumor. PCr/ATP, PCr/Pi, Pi/ATP, PDE/ATP, PDE/PCr, and PDE/Pi were lower in tumor voxels compared to the healthy-looking brain voxels, while PME/PDE, PME/ATP, and PME/PCr were increased in tumor voxels [67]. Similar results were found by Hmilicova et al. [68]. Lower PCR/Pi, PDE/ATP, and higher pH were also found in a study by Maintz et al. [69], while Bulakbasi et al. found similar results regarding the intracellular pH, Mg levels, and PCr/ATP, PCr/Pi, PME/ATP, and PDE/ATP ratios [70]. Hattingen et al. found decreased PCr/Pi and increased Pi/ATP in tumor areas in patients with recurrent GBM treated with bevacicumab [71]. This implied antitumoral effects of bevacicumab and impaired oxidative energy metabolism in GBM treated with this drug. Similar to the findings of Walchhofer et al., the group around Ha et al. found increased PME/PDE and PME/PCr ratios in tumor areas [72]. Kamble et al. also found similar results regarding to the PCr/ATP, PCR/Pi, and PME/PDE ratios [73].

## 6. Limitations

### 6.1. 1H-MRS

1H-MRS is a promising method, but encounters several limitations, e.g., lesions near the bone due to magnetic susceptibility artifacts [58]. In addition, spectroscopy can accurately distinguish tissues containing pure RN from pure recurrence. However, metabolite values are averaged within the studied voxel in monovoxel studies, and consequently the co-existence of both effects constitute a major challenge in the interpretation of resulting spectra [57]. In contrast, multivoxel spectroscopy enables a more thorough examination of metabolic changes.

A further consideration in 1H-MRS is voxel positioning: tumor recurrence may further be more accurately depicted by means of spectroscopy in areas which do not enhance, as well as in the adjacent white matter, as typical tumor spectrum profiles are often depicted in these areas [6]. A promising upcoming method is 3D echoplanar spectroscopic imaging, where a large volume can be analyzed with greater resolution [74].

Some pros and cons of using 1H-MRS in clinical practice is given in Table 3.

### 6.2. 31P-MRS

The future clinical benefit of 31P-MRS still has to be verified with clinical and confirmatory studies. The clinical application of this method has remained limited until today due to several factors.

First of all, the restricted availability of the coils and the technique renders clinical establishment difficult, especially in non-specialized centers.

Due to varying MRI technique and field strength used in the published studies reliable comparison of data is challenging. Regardless of these limitations, standardization is also lacking due to the limited number of studies or case series available for direct comparison and confirmation of published findings. The small sample size of published studies plus the heterogeneity of tumor patients render it difficult to draw generally accepted conclusions for clinical application. Thus, confirmatory studies involving this imaging technique are needed to render the interpretation of 31p-MRS results into a clinical applicable routine procedure [32].

Furthermore, the effect of “voxel bleeding” due to a poor point spread function is an omnipresent problem in MRS. This effect can be minimized by choosing voxels in which the tissue to investigate is present in at least two-thirds of the voxel. This approach has been shown to be of high value to retrieve significant and reliable results [66].

Another important reason for the lack of clinical application has been the lack of mean normal values in healthy controls as a comparison for the use of 31P-MRS. Particularly the knowledge of differences in the brain depending on sex, age and brain region are crucial for the interpretation of derived results. In a large study, it was found that ATP-resynthesis, ATP-hydrolysis and energy demand vary between brain regions, age and sex. Therefore, these parameters also have to be taken into account when investigating cerebral energy metabolism under pathological conditions. This renders the interpretation of 31P-MRS even more challenging [33].

31P-MRS studies in neurooncology also have several limitations inherent to the aggressive nature of the investigated tumors and the rapidly deteriorating health condition of effected patients. As 31P-MRS scans are time-intensive and, this examination can be intolerable for some GBM patients, especially in later disease stages. Consequently, published 31P-MRS studies usually include a small number of patients for whom scans were available, as follow up examinations experience a large number of dropouts. Consequently, large cohort studies are sparse and consequently confirmatory studies for more recent findings still lacking. Although state-of-the-art therapy, GBM bears a short survival time, thus rendering this limitation difficult to overcome [66].

A particular challenge in tumor imaging are the poor signal-to-noise ratios and, therefore, a noisier baseline of the spectra in tumor regions, which might potentially lead to an exclusion of two metabolites from analyses [67].

Additionally, the energy and membrane metabolism is modified in the entire brain of patients with GBM, even in “normal-appearing” brain areas and the contralateral hemisphere. Although GBM is an aggressive infiltratory process, this might also be explained by therapeutic effects on pre-therapeutic presumably healthy brain tissue. Assuming changes under therapy, the distinction between therapeutic effects and tumor progress is a challenging endeavor. This is further complicated by the fact that observed changes are also dependent on the therapeutic success [66].

31P-MRS, when interpreted along with other clinical and imaging parameters, constitutes an additional imaging biomarker for both outcome measurement or treatment response. This is of potential interest in radiomics studies, and could potentially enable a more reliable and reproducible non-invasive diagnosis and more individualized treatment planning.

At the moment this method is experimental and still has to be implemented in clinical investigation [33].

## 7. Conclusions

Proton MR-spectroscopy is a valuable addition to the conventional MR sequences in diagnosis, and especially in follow-up of patients with high-grade glioma. Furthermore, we believe that the spectroscopy techniques can be a valuable tool in research of glioma, and that the 1H-, but also 31P- and other types of spectroscopy can provide us with a new insight in the metabolism of a glioma tissue in vivo.

## Figures and Tables

**Figure 1 cancers-14-03197-f001:**
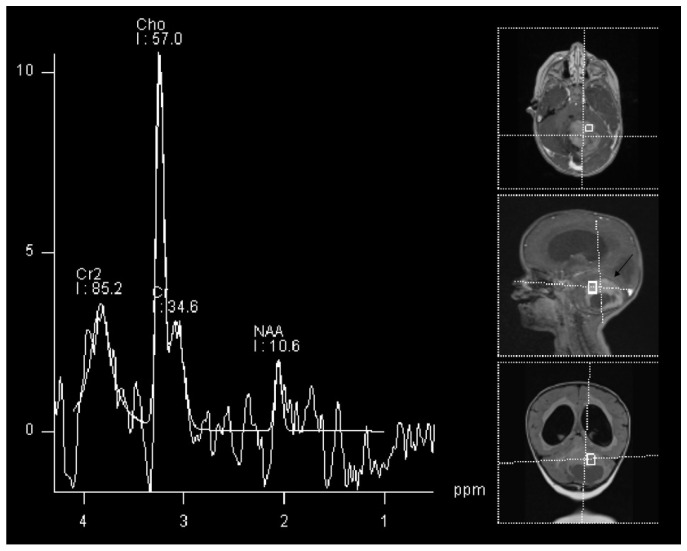
1H-MRS in a patient (2 Y) with pilocitic astrocytoma (black arrow). Significantly increased Cho and decreased NAA and Cr are observed.

**Figure 2 cancers-14-03197-f002:**
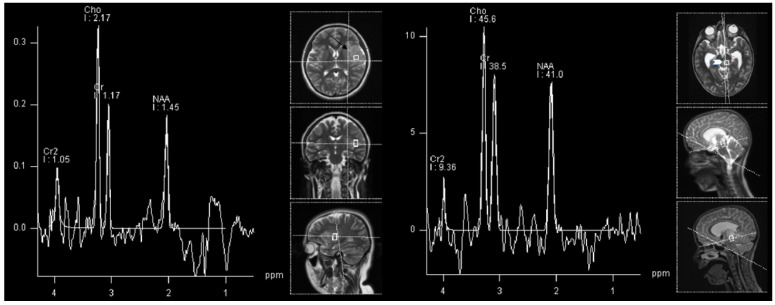
1H-MRS in a patient (15 Y) with astrocytoma grade 3 (**left**, black arrow) and in a patient (8 Y) with astrocytoma WHO grade 2 (**right**, white arrowhead). Similar spectra are present in both grade 3 and grade 2 astrocytoma with increased Cho and decreased NAA, with the absence of Cr decrease.

**Figure 3 cancers-14-03197-f003:**
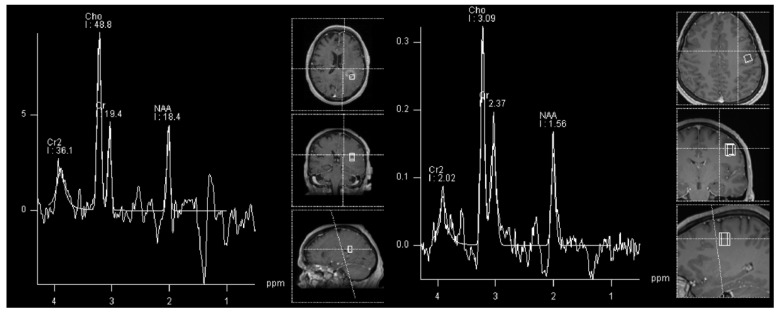
Similar 1H-MRS spectra can be appreciated in an adult patient with glioblastoma (**left**, black arrow) and in a patient with astrocytoma WHO grade 2 (**right**, white arrowhead) with increased NAA and decreased Cr and Cho.

**Figure 4 cancers-14-03197-f004:**
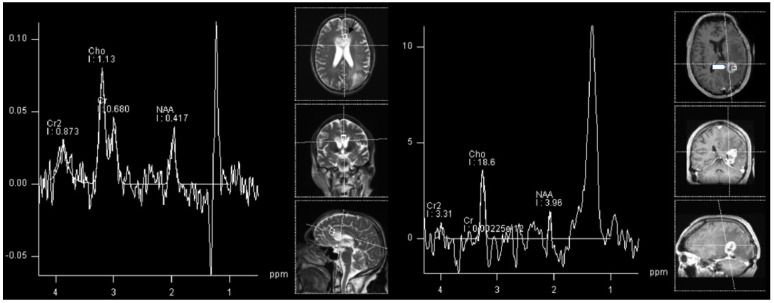
1H-MRS in an adult patient with glioblastoma progression (**left**, black arrow) and in a patient with signs of both progression and radiation necrosis (**right**, white arrowhead). Cho/NAA ratio is under proposed cut-off for TP in progressive GBM.

**Table 1 cancers-14-03197-t001:** Classification of gliomas according to the new 2021 WHO Classification [10]. The abbreviations are given in the Abbreviations section.

	WHO Grade	Most Common Molecular Features
**Circumstribed gliomas**		
Pilocytic astrocytoma	1	KIAA1549-BRAF [11]
High-grade astrocytoma with piloid features	new subtype	specific DNA-methylation profile [12]
Pleomorphic xanthoastrocytoma	2, 3	BRAF [13]
Subependymal giant cell astrocytoma	1	TSC1, TSC2 [14]
Chordoid glioma	2	PRKCA [15]
Astroblastoma, MN1-altered	new subtype	MN1 [16]
**Pediatric diffuse low grade gliomas**		
Diffuse astrocytoma, MYB- or MYBL1-altered	1	MYB, MYBL1 [17]
Angiocentric glioma	1	MYB [10]
Polymorphous low-grade neuroepithelial tumor of the young	1	PLNTYs, BRAF, FGFR [18]
Diffuse low-grade glioma, MAPK pathway-altered	new subtype	FGFR1, BRAF [19]
**Pediatric-type diffuse high-grade gliomas**		
Diffuse midline glioma, H3 K27-altered	4	H3 K27 [20]
Diffuse hemispheric glioma, H3 G34-mutant	4	H3F3A (G34R/V) [21], GFAP [22], p53 [23]
Diffuse pediatric-type high-grade glioma, H3- and IDH-wt	4	IDH-wt, H3-wt, MYCN, PDGFRA [24]
Infant-type hemispheric glioma	new subtype	NTRK, ALK, ROS1, MET [25]
**Adult-type diffuse gliomas**		
Astrocytoma, IDH-mutant	2, 3, 4	IDH1, IDH2 [26], ATRX [27]
Oligodendroglioma, IDH-mutant, and 1p/19q-codeleted	2, 3	IDH [28], 1p19q-codeletion, ATRX, p53 [29]
Glioblastoma, IDH-wt	4	no IDH mutation (IDH-wt), ATRX, TERT [30]

**Table 2 cancers-14-03197-t002:** Used parameters for 31P-MRS acquisition.

Matrix	8 × 8 × 8
Field of view	240 × 240 × 200 mm ${}^{3}$
Voxel size	30 × 30 mm ${}^{2}$
Slice thickness	25 mm
Repetition time	2000 ms
Echo time	2.3 ms
Flip angle	90${}^{\circ }$

**Table 3 cancers-14-03197-t003:** Pros and cons of including 1H-MRS in a clinical routine of glioma imaging.

Pros	Cons
Important information about the nature of the lesions	Hardware, software and know-how considerations-cost
More accurate follow-up	Relatively time costly and artifact-prone sequence

## Data Availability

Not applicable.

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
