# Peer review of "Magnetic Resonance Spectroscopy in Diagnosis and Follow-Up of Gliomas: State-of-the-Art"

_cancers, 2022, doi:10.3390/cancers14133197_

Round 1

Reviewer 1 Report

The aim of the paper was to investigate the clinical usefulness of magnetic resonance spectroscopy (MRS) in patients with gliomas, analyzing the topic regarding the new 2021 WHO classification of those tumors. MRS is a well-known addition to conventional MR diagnosis in gliomas, both before treatment and in the follow-up. The authors researched MRS value in pediatric and adult types of glioma.

Since based on molecular features WHO Classification of Tumors of Central Nervous System (2021) introduced many changes for glial brain tumors – it is utterly important to review the value of commonly available 1H-MRS and novelty 31P-MRS in classifying the tumor subtype before the removal and in distinguishing radiation necrosis, pseudoprogression and tumor progression in the follow-up. The authors chose to review a very significant state-of-the-art of MRS in glioma patients and completed the topic of this review.

The manuscript is clear, relevant for the filed and presented in a well-structured manner. The figures and table properly show the data and are easy to interpret. The cited references are relevant. The conclusions are coherent and well supported by the listed citations.

My specific comments:

1.     Abbreviations and acronyms used in “Molecular features” in Table 1 have not been explained.

2.     For 31P-MRS specific ratios are not explained in the article (lines 226-227): PCr/ATP, PCr/Pi, Pi/ATP, PDE/ATP, PDE/PCr, PDE/Pi, PME/PDE, PME/ATP, PME/PCr.

Reviewer 2 Report

The authors aim to review the technical aspects of MRS and the clinical valuable it provides to distinguish between the glioma grading in both adult and pediatric patients and especially how to differentiate between TP, RN and PsP in GBM patients.  Furthermore, it reviews the current state and future potential of 31P-MRS. Previous reviews of MRS incl. for glioma grading has been done before, however, this paper provides an excellent, clear, and up-to-date overview of MRS use in glioma with the novelty of WHO 2021 classification and how MRS can potentially be used in tumors with various molecular compositions.   Below are some minor comments and suggestions:    
  1. Technical overview
    65, 75
    Section 3.1 and 3.2 describes the basics of how the pediatric and adult patients are positioned and observed during the MRI. This section can be shortened.
  2. Diagnosis and 1H-MRS
    106 “.. most common glioma in pediatric population is pilocytic astrocytoma, [ref11] Pediatric Brain Tumor Genetics: What Radiologists Need to Know”

    Change the citation to include the primary (updated) source “CBTRUS Statistical Report: Primary Brain and Other Central Nervous System Tumors Diagnosed in the United States in 2014-2018”

    113 “… high-grade astrocytomas and GBMS are most common high-grade gliomas in children[ref13], Fangusaro, J. Pediatric High Grade Glioma: a Review and Update on Tumor Clinical Characteristics and Biology

    Change the reference to an up-to-date ref, e.g. “CBTRUS Statistical Report: Primary Brain and Other Central Nervous System Tumors Diagnosed in the United States in 2014-2018”

  3. Low-grade vs. High-grade
    128
    ” In gliomas, general rule is that the NAA and Cr decrease, and other metabolites increase with higher tumor grade. However, as shown in Figure 3, low-grade gliomas can also have similar spectra as high-grades (but rarely vice-versa).”

    Please insert citations to support the statement.

     It is stated that preoperative grading of gliomas can be difficult based on conventional MRI alone and that additional information can be obtained with MRS, which begs the question how useful is MRS in preoperative grading? Is there any evidence on the specificity and sensitivity of adding preoperative MRS compared to conventional MRI and perfusion?

    How common is it that low grade can have same the spectra as high grade?

  4. Limitations
MRS is not routinely used in the clinical setting. What are some of the limitations in the clinical setting to implementing MRS?

Reviewer 3 Report

To editors and authors

Magnetic Resonance Spectroscopy in Diagnosis and Follow-up of Gliomas: State-of-the-Art

This is a very interesting manuscript that should be considered for publication in CANCERS after some revisions below.

1) Please recheck and revise cautiously citation and references as MDPI format.

2) In each imaging, please add icon like arrow or arrowhead to point out the lesions.

3) I believe that imaging part lacked a very updated reference related to this manuscript strictly. PMID: 32588986 and PMID: 32366451

4) It is better to have histopathological images.

Sincerely

Reviewer 4 Report

The review addresses an important aspect for the diagnosis of CNS cancers. Compared to the reviews in the literature, the comparison between adult and pediatric is prestigious.

1)      First sentence of introduction ….Patients benefit from 9 an early diagnosis and precise follow up…. Please specify and inserted recent data, analysis or review.

2)      please insert a short paragraph comparing pros and cons of MRI versus proton-MRS, possibly with a table.
